# (GIGA)byte

DATA RELEASE

# A reference genome for the critically endangered woylie, *Bettongia penicillata ogilbyi*

Emma Peel[1], Luke Silver[1], Parice Brandies[1], Carolyn J. Hogg[1] and Katherine Belov[1,*]

1 School of Life and Environmental Sciences, The University of Sydney, Sydney, New South Wales, Australia

## ABSTRACT

Biodiversity is declining globally, and Australia has one of the worst extinction records for mammals. The development of sequencing technologies means that genomic approaches are now available as important tools for wildlife conservation and management. Despite this, genome sequences are available for only 5% of threatened Australian species. Here we report the first reference genome for the woylie (*Bettongia penicillata ogilbyi*), a critically endangered marsupial from Western Australia, and the first genome within the Potoroidae family. The woylie reference genome was generated using Pacific Biosciences HiFi long-reads, resulting in a 3.39 Gbp assembly with a scaffold N50 of 6.49 Mbp and 86.5% complete mammalian BUSCOs. Assembly of a global transcriptome from pouch skin, tongue, heart and blood RNA-seq reads was used to guide annotation with Fgenesh++, resulting in the annotation of 24,655 genes. The woylie reference genome is a valuable resource for conservation, management and investigations into disease-induced decline of this critically endangered marsupial.

**Subjects** Genetics and Genomics, Animal Genetics, Genetics

**Submitted:** 15 September 2021

\* Corresponding author. E-mail: kathy.belov@sydney.edu.au

Preprint submitted at https: //doi.org/10.1101/2021.12.07.471656

## DATA DESCRIPTION

### Background and context

Globally, we are experiencing a biodiversity crisis, as more than 1 million species currently face extinction [1]. Australia is one of 17 megadiverse countries [2], and has a high level of endemism, with 87% of mammals, 94% of frogs and 93% of reptiles being endemic to Australia [3]. Despite this, Australia has one of the worst mammal extinction rates in the world, with over 10% of endemic terrestrial mammals driven to extinction within the past 200 years [4]. The International Union for Conservation of Nature (IUCN) currently lists over 1000 Australian animals as critically endangered, endangered or vulnerable as of August 2021 [5], yet genome sequences are only available for 5% of these species.

Reference genomes are a valuable conservation tool that allow researchers and managers to answer vital biological questions and inform management policy [6]. Development of new sequencing technologies and their subsequent decrease in cost allows the genomes of threatened species to be sequenced [7–9]. Individual research groups can now assemble highly contiguous genomes, which can then be used to answer various biological, evolutionary and conservation questions [10–12].

The woylie, or brush-tailed bettong (*Bettongia penicillata ogilbyi*, NCBI:txid881300), is a small marsupial of the Potoroidae family (Figure 1) [13]. Marsupials are one of three

**Figure 1.** Woylie *Bettongia penicillata ogilbyi* from Western Australia. Photo credit: Sabrina Trocini.

lineages of mammals, the others being eutherians (e.g. humans and mice) and monotremes (e.g. platypus and echidna). Since their divergence from eutherian mammals around 156 million years ago, marsupials have evolved into over 300 species, most of which are endemic to Australia [14]. Marsupials differ to other mammals in several ways; the most prominent is their pouch. After a short gestation of only 20 days, woylies give birth to altricial young, which develop within the pouch for 100 days [13]. The complex milk profile of the mother provides nutrient and immune support throughout pouch life [15]. Similar to wallabies and kangaroos, woylies undergo embryonic diapause, whereby embryonic development is suspended by the suckling pouch young, and resumes when the young exits the pouch [13]. Woylies are ecosystem engineers, with individuals displacing on average 4.8 tonnes of soil per year [16]. This bioturbation is essential for ecosystem health and function, as it activates and disperses the mycorrhizal fungi comprising a large portion of their diet [17, 18], alters soil nutrient composition and water penetration [19], and aids seed dispersal [20].

Historically, woylies inhabited much of central and southern Australia; however, populations have contracted to 1% of their former range owing to habitat loss and fragmentation [4, 21]. Between 1999 and 2006, woylie populations significantly declined by more than 90% [22], resulting in their current IUCN listing as critically endangered [5]. The decline is thought to be caused by a combination of predation by introduced feral cats (*Felis catus*) and foxes (*Vulpes vulpes*), and an unknown disease linked to parasites such as trypanosomes [23–25]. Currently, there are two remaining indigenous populations in the Upper Warren and Dryandra regions of Western Australia [22]. In response to the decline, several translocations and reintroductions have been conducted within Western Australia (WA), South Australia (SA) and New South Wales (NSW), guided by genetic assessments of

diversity and population structure using microsatellite [26, 27] and genomic-based methods [28].

In this study, we present the first *de novo* reference genome assembly of the woylie, as well as four tissue transcriptomes. This is the first genome sequenced within the Potoroidae family, which will be a valuable tool for investigating disease-induced decline, basic biology, and to aid conservation.

## METHODS

### Sample collection and sequencing

Spleen, heart, kidney and tongue were opportunistically sampled from a single wild female woylie (woy01), as well as pouch skin from a second wild female woylie (woy02), both of which died by vehicle strike at Manjimup, Western Australia (WA) in 2018. In addition, 500 µL of peripheral blood was collected into RNAprotect Animal Blood tubes (Qiagen) from a third wild male woylie (woy03) from Balban, WA, in 2018 during routine trapping and health examinations. All samples were collected under the Western Australian Government Department of Biodiversity, Conservation and Attractions animal ethics 2018-22F and scientific licence number NSW DPIE SL101204.

High-molecular-weight (HMW) DNA was extracted from woy01 kidney using the Nanobind Tissue Big DNA kit (Circulomics), and quality assessed using the NanoDrop 6000 with an A260/280 of 1.91 and A260/230 of 2.37. HMW DNA was submitted to the Australian Genome Research Facility (Brisbane) for Pacific Biosciences (PacBio) HiFi sequencing. Briefly, the DNA was sheared using the Megaruptor2 kit to generate 15 to 20-Kbp (kilobase pair) fragments. The BluePippin SMRTbell Library Kit was then used to select DNA fragments longer than 15 Kbp, which were used as input to the SMRTbell Express Template Prep Kit 2.0. The resulting PacBio HiFi SMRTbell libraries were sequenced across two single-molecule real-time (SMRT) cells on the PacBio Sequel II. This resulted in 37 Gbp (gigabase pairs) of raw data.

For 10X Chromium linked-read sequencing, HMW DNA was extracted from 25 mg of woy01 spleen using the MagAttract HMW DNA kit (Qiagen), and quality was assessed using the NanoDrop 6000 with an A260/280 and A260/230 of 1.8–2.3. HMW DNA was submitted to the Ramaciotti Centre for Genomics (UNSW) for 10X Chromium genomics library preparation, and 150-bp (base pair) paired-end (PE) reads were sequenced on an Illumina NovaSeq 6000 S1 flowcell. This generated 137 Gbp of raw data.

Total RNA was extracted from 25 mg of woy01 tongue and heart, woy02 pouch skin, using the RNeasy Plus Mini Kit (Qiagen). In addition, total RNA was extracted from 500 µL of woy03 peripheral blood using the RNAprotect Animal Blood Kit (Qiagen). In all extractions, contaminating DNA was removed through on-column digestion using the RNase-free DNase I set (Qiagen). RNA purity was assessed using the NanoDrop 6000, with all samples displaying an A260/280 and A260/230 of 1.9–2.2. RNA concentration and integrity were measured using an RNA Nano 6000 chip (Agilent Technologies), with all samples displaying an RNA integrity number (RIN) greater than 7. Total RNA was submitted to the Ramaciotti Centre for Genomics (University of New South Wales) for TruSeq mRNA library preparation. All tissue libraries were sequenced as 150-bp PE reads across one lane of an S1 flowcell on the NovaSeq 6000, while the blood library was sequenced as 150-bp PE reads across an SP flowcell on the NovaSeq 6000. This resulted in 23–27 GB (gigabytes) raw data per sample. All genomic and transcriptomic data generated in this study are summarised in Table 1.



**Table 1.** Summary of sequencing data generated in this study.

| Sequencing platform | Data type | Individual/tissue | Raw data (GB) |
|---|---|---|---|
| PacBio Sequel II | HiFi reads | woy01 kidney | 37 |
| Illumina NovaSeq6000 | 10x linked-reads | woy01 spleen | 137 |
| Illumina NovaSeq6000 | RNA-seq reads | woy01 tongue and heart woy02 pouch skin woy03 blood | 23–27 per sample |

## Genome assembly and annotation

Raw sequencing data was quality checked using SMRT Link v9.0.0.92188 [29] and fastQC v0.11.8 (RRID:SCR_014583) [30]. Sequences were assembled d*e novo* using Improved Phased Assembler (IPA) v1.1.2 [31] with default parameters on the Nimbus cloud service provided by the Pawsey Supercomputing Centre (virtual machine – 64 vCPUs; 256 GB RAM; 3 TB Storage). Assembly statistics were obtained using BBmap v37.98 (RRID:SCR_016965) [32] and assembly completeness assessed using BUSCO (Benchmarking Universal Single-Copy Orthologs) v4.0.6 and v3.1.0 (RRID:SCR_015008) [33]. The assembly was then filtered to remove duplicate haplotigs using purgedups v1.0.1 [34, 35]. The *de novo* assembly was then scaffolded with 10x linked reads using 10x Genomics Long Ranger v2.2.2 (RRID:SCR_018925) and arcs v1.1.1 [36]. Additional gap filling was performed with the HiFi data using PBJelly v14.1 (RRID:SCR_012091) [9]. The genome was polished with Pilon v1.20 (RRID:SCR_014731) [37] by converting the 10x linked reads to standard Illumina reads by removing 10x adapter sequences. For annotation, a custom repeat database was generated using RepeatModeler v2.0.1 (RRID:SCR_015027) [38], then RepeatMasker v4.0.6 (RRID:SCR_012954) [39] was used to mask repeats, excluding low complexity regions and simple repeats. Functional completeness was assessed using BUSCO v4.0.6 and v3.1.0 (RRID:SCR_015008) against the mammalian database [40]. Genome annotation was then performed using Fgenesh++ v7.2.2 (RRID:SCR_018928) [41] with general mammalian pipeline parameters and an optimised gene-finding matrix from Tasmanian devils (*Sarcophilus harrisii*). Transcripts with the longest open reading frame for each predicted gene were extracted from the global transcriptome for mRNA-based gene predictions. The National Center for Biotechnology Information (NCBI) non-redundant protein database was used for protein-based gene predictions [42]. Statistics for protein-coding genes were calculated using genestats [43].

## Transcriptome assembly and annotation

Raw RNA-seq data was quality checked using fastQC v0.11.8 (RRID:SCR_014583) [30], then and length and quality trimmed using Trimmomatic v0.38 (RRID:SCR_011848) [44] with the following flags: ILLUMINACLIP:TruSeq3-PE.fa:2:30:10 SLIDINGWINDOW:4:5 LEADING:5 TRAILING:5 MINLEN:25. Illumina TruSeq sequencing adapters were removed from the dataset (ILLUMINACLIP:TruSeq3-PE.fa:2:30:10), as well as reads shorter than 25 bp (MINLEN:25). Reads were quality trimmed and removed where the average quality score fell below 5 within a 4-bp sliding window (SLIDINGWINDOW:4:5), as well as at the 5′ (LEADING:5) and 3′ (TRAILING:5) end of the read. Over 99.7% of reads were retained for all datasets post-trimming.

A global transcriptome for the woylie was produced by aligning trimmed reads from four tissues (heart and tongue from woy01, pouch skin from woy02 and whole blood from woy03) to the final genome assembly using hisat2 v2.1.0 (RRID:SCR_015530) [45] using

default parameters. This produced sam files, which were then sorted using SAMTOOLS v1.6 (RRID:SCR_002105) [46] to produce bam files for each tissue. StringTie v2.1.4 (RRID:SCR_002105) [47] was used to produce gtf files for each tissue. Tama merge [48, 49] was used to merge aligned reads from each tissue to a global transcriptome with a 5′ threshold of 3 and 3′ threshold of 500. Transcripts were then removed if they were only present in one tissue sample or had a fragments per kilobase of transcript per million (FPKM) of less than 0.1. CPC2 [50, 51] was used to predict whether a transcript was a coding gene, and to filter out transcripts with low expression. TransDecoder v2.0.1 (RRID:SCR_017647) [52] was then used to determine coding regions and open reading frames within the transcripts. The number of full-length protein coding genes was determined by using BLAST (Basic Local Alignment Search Tool) [53] to identify top hits of the full-length TransDecoder-predicted proteins against the Swiss-Prot non-redundant database, available at UniProt [54]. Functional completeness was assessed using BUSCO v4.0.6 and v3.1.0 (RRID:SCR_015008) against the mammalian database [40]. To determine read representation and generate transcript counts, trimmed reads were mapped back to the global transcriptome assembly using bowtie2 v2.4.4 (RRID:SCR_005476) [55] with default parameters, except a maximum of 20 distinct valid alignments for each read. These alignments were used as input to Salmon v1.4.0 [56] to generate transcript per million (TPM) counts for each tissue. TransDecoder-predicted proteins expressed in the pouch skin with hits to Swiss-Prot (e-value threshold of $e^{-5}$) were used as input to Panther (RRID:SCR_004869) [57], where they were assigned Gene Ontology (GO) slim terms under the Biological Process category.

## RESULTS AND DISCUSSION

### Genome

The *de novo* woylie genome assembly was 3.39 Gbp in size, like other marsupial genomes (Table 2). The genome was assembled into just over 1000 scaffolds, with a scaffold N50 of 6.94 Mbp, and is more contiguous than the tammar wallaby genome, the closest relative with an available genome [58]. Gaps made up 0.40% of the genome; fewer than antechinus (*Antechinus stuartii*) (2.75%) [59] but higher than koala (*Phascolarctos cinereus*) (0.1%), which is not surprising given the numerous sequencing technologies used to generate the high-quality koala genome assembly [60]. The high scaffold N50 for the woylie genome relative to other assembly statistics is likely to be associated with the presence of long contigs derived from the long HiFi reads. The longest contig in the assembly was 12 Mbp, with eight contigs longer than 5 Mbp. Following scaffolding with 10x linked-reads, three scaffolds in the assembly were longer than 25 Mbp (longest 35.66 Mbp) and 72 were longer than 10 Mbp. Despite this, the high scaffold N50 may be attributed to scaffolding error. The genome presented here is a high-quality draft assembly and provides a basis for future improvement.

Repeat elements comprised 53.05% of the woylie genome, similar to the tammar wallaby (*Notamacropus eugenii*) (52.8%) [58], but higher than antechinus (44.82%) [59] and koala (47.5%) [60]. Repeat families numbering 1184 were identified in the woylie genome, with long interspersed nuclear elements (LINEs) and short interspersed nuclear elements (SINEs) being the most numerous (Table 3), as in other marsupial genomes [59, 60, 64]. Retrotransposon-like elements (RTE) were also identified in the woylie genome, as observed in other marsupial genomes and some mammals such as ruminants [59, 65, 66].

**Table 2.** Assembly metrics for the woylie genome, compared with other published marsupial genomes.

| Metric | Woylie (present study) | Koala [60–62] | Antechinus [59] | Tasmanian devil [63] | Tammar wallaby [58] |
|---|---|---|---|---|---|
| Year | 2021 | 2018 | 2020 | 2012 | 2011 |
| Genome size (Gbp) | 3.39 | 3.19 | 3.31 | 3.17 | 2.7 |
| No. scaffolds | 1116 | 1318 | 30,876 | 237,291 | 277,711 |
| No. contigs | 3016 | 1935 | 106,199 | 35,974 | 1,174,382 |
| Scaffold N50 (Mbp) | 6.94 | 480.11 | 72.7 | 1.8 | 0.036 |
| Contig N50 (Mbp) | 1.99 | 11.4 | 0.08 | 0.02 | 0.002 |
| GC (%) | 38.64 | 39.05 | 36.20 | 36.04 | 38.8 |

**Table 3.** Repeats identified in the woylie genome.

| Repeat type | Number | Length (bp) | Sequence masked Percentage (%) |
|---|---|---|---|
| SINE | | | |
| ALUs | 16,376 | 3,542,886 | 0.10 |
| MIRs | 2,012,972 | 349,844,958 | 10.31 |
| LINE | | | |
| LINE1 | 1,054,578 | 588,827,450 | 17.35 |
| LINE2 | 1,261,114 | 282,854,529 | 8.34 |
| CR1 | 553,626 | 102,521,146 | 3.02 |
| LTR | | | |
| ERVL | 16,900 | 7,241,092 | 0.21 |
| ERV1 | 22,176 | 7,069,536 | 0.21 |
| ERV2 | 22,364 | 11,265,604 | 0.33 |
| DNA elements | | | |
| hAT-Charlie | 141,744 | 23,630,550 | 0.70 |
| TcMar-Tigger | 36,287 | 8,389,535 | 0.25 |
| Other | | | |
| Unclassified | 1,017,337 | 231,493,940 | 6.82 |
| Satellite DNA | 29,228 | 4,458,406 | 0.13 |
| Small RNA | 615 | 53,205 | 0.00 |

Interestingly, primate-specific ALU repeats were identified in the woylie genome. ALU repeats are a type of SINE that comprise over 10% of the human genome and are involved in genome evolution and disease [67]. These repeat elements contributed only 0.10% to the woylie genome, and were also identified in the antechinus genome (0.04%) [59]. As this may represent an inaccurate repeat annotation, further work is required to confirm the presence of ALU repeats in marsupials.

Fgenesh++ predicted 41,868 genes in the woylie genome, of which 24,655 had BLAST hits to eukaryote genes in the NCBI non-redundant database. Of these 24,655 genes, 15,904 were supported by mRNA evidence, and 1309 by protein evidence. This is higher than the number of protein-coding genes annotated by NCBI in the koala genome (20,103) [60] and Tasmanian devil (20,053) [63] (Table 4). The higher number of genes annotated in the woylie genome is likely due to incomplete RNA-seq evidence, and hence gene models, used for gene prediction by Fgnesh++. In addition, fragmentation of the genome causes fragmentation of gene sequences, which can result in an overinflated gene count [68, 69]. Statistics for protein-coding genes annotated within the woylie genome also reflected deficiencies in gene models, as mean gene and exon length, and mean exon number per gene differed to the NCBI annotation of the koala and devil genome (Table 4). The NCBI annotation pipeline uses mRNA and protein evidence from multiple public scientific

**Table 4.** Statistics for protein-coding genes annotated in the woylie genome compared to NCBI annotations of the koala and Tasmanian devil genomes. Accession numbers provided.

| Parameter | Woylie (this study) | Koala (GCF_002099425.1) | Tasmanian devil (GCF_000189315.1) |
|---|---|---|---|
| No. protein-coding genes | 24,655 | 20,103 | 20,053 |
| Mean gene length (bp) | 24,136.6 | 55,640 | 45,875 |
| Mean exon length (bp) | 550 | 291 | 269 |
| Mean intron length (bp) | 8559 | 7261 | 5936 |
| Mean exon number per gene | 3.58 | 11.11 | 10.36 |

databases for gene prediction, resulting in an annotated gene number that more closely resembles humans (~20,000) [70].

## Transcriptome

The woylie global transcriptome assembly of four tissues (blood, heart, pouch skin and tongue) contained 145,939 transcripts, with an average transcript length of 7739 bp and transcript N50 of 15,469 bp. TransDecoder predicted 151,147 coding regions within the global transcriptome, of which 74% were complete (contained a start and stop codon) and 75.4% had BLAST hits to the Swiss-Prot non-redundant database. The number of TransDecoder proteins expressed in the pouch skin with BLASTp hits to Swiss-Prot was 89,707, of which 21,856 represented unique Swiss-Prot entries. Of these, 17,588 were assigned GO-slim terms under the biological process (BP) category, with cellular processes (29.8%), metabolic processes (18.9%) and biological regulation (16.7%) being the most common GO terms (Figure 2). Of the GO-slim terms under the BP category, 1.4% were involved in immune system processes, such as immune cell development, activation, and antigen processing (Figure 2).

As this study presents the first marsupial pouch skin transcriptome, and given the important protective role of the pouch, we investigated the top transcripts expressed in this tissue. Of the top 10 transcripts expressed in the pouch skin with hits to Swiss-Prot, four were involved in innate immune defence (Figure 3). The most highly abundant pouch skin transcript was lysozyme C. Lysozyme is an ancient antimicrobial enzyme conserved throughout evolution [71], which degrades the peptidoglycan layer of bacterial cell membranes. Calcium binding proteins of the S100 family, such as S100-A9 and S100A15A, and surfactant-associated protein D (SP-D), were also highly expressed in the pouch skin. These proteins are involved in innate immunity, and are chemotactic [72], antimicrobial [73–75] and modulate inflammation [76]. Marsupial young, including the woylie, are born immunologically naïve without mature immune tissues or cells [77]. The abundance of innate immune proteins in the pouch skin transcriptome highlights the importance of the pouch in protecting naïve young during development. As adaptive immunity does not completely mature until 100 days after birth in some species, the young rely on passive immunity from the milk, rapid development of the innate immune system, and antimicrobial compounds from the pouch for protection against pathogens [78–82]. Antimicrobial compounds expressed in the pouch skin likely contribute to changes in the pouch microbiome throughout lactation in marsupials [83, 84], and may selectively eliminate pathogens via direct antibacterial activity [79, 85, 86].

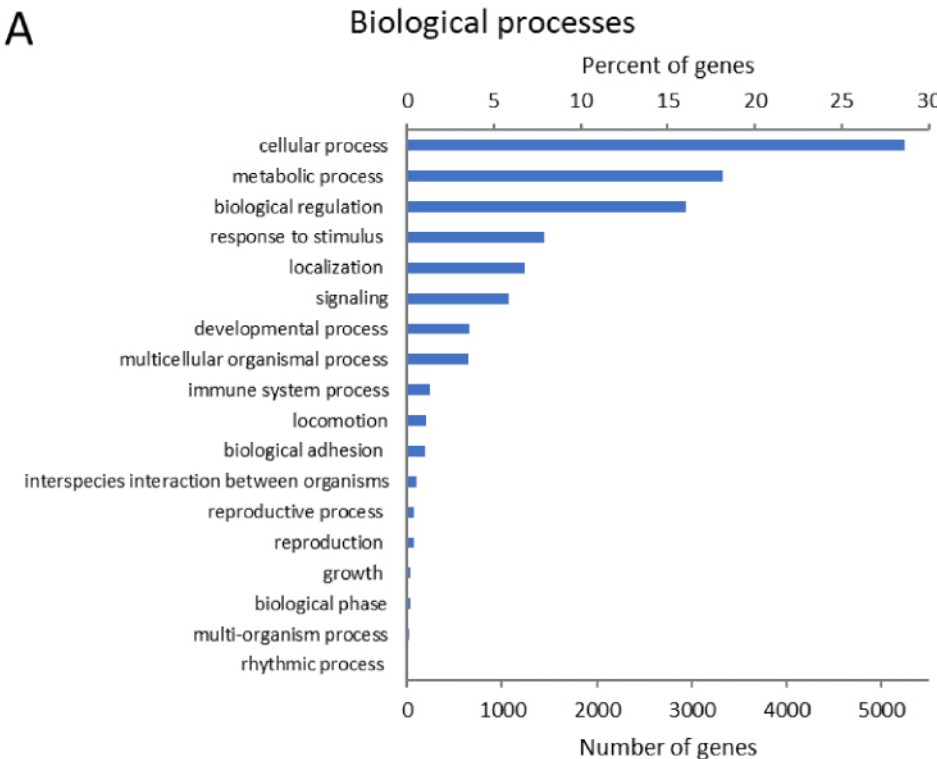

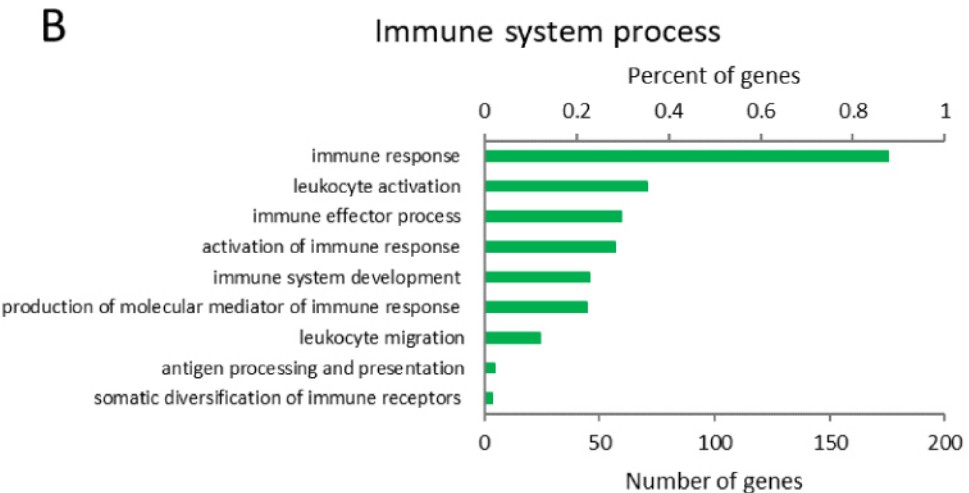

**Figure 2.** Top GO terms assigned to proteins expressed in the pouch skin with Swiss-Prot hits. (A) Proteins identified under the biological process category and (B) the immune system process term. Both the number of genes and percentage of total genes assigned to each category are provided.

## DATA VALIDATION AND QUALITY CONTROL

BUSCO was used to assess functional completeness by searching for complete single-copy gene orthologs within the genome assembly, Fgenesh++ predicted proteins and the global transcriptome assembly. The genome contained 86.5% of complete mammalian BUSCOv4

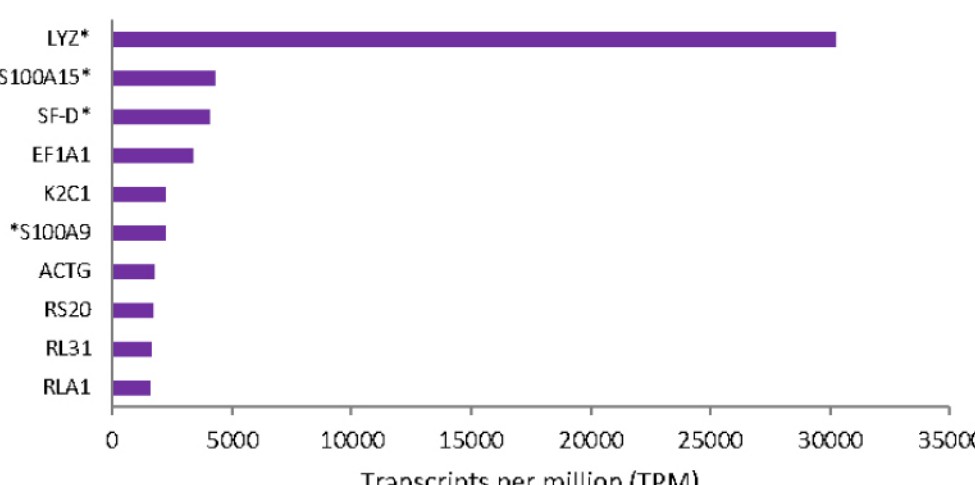

**Figure 3.** Transcript per million (TPM) counts of the top 10 proteins expressed in the pouch skin with hits to Swiss-Prot. Proteins involved in innate immunity are indicated by *.

genes, comparable to other marsupial genomes [58–60, 63, 87]. Fgenesh++-predicted proteins and the global transcriptome also displayed a high level of completeness, with 80.4% and 80.8% of complete mammalian BUSCOv4 identified, respectively.

High mapping rates were observed for both the genome and global transcriptome assemblies, indicating high sequencing accuracy and low contaminating DNA. 99.8% of HiFi reads and 88.4% of 10x Chromium Illumina reads mapped to the genome assembly. Similarly, 80.25% (blood), 70.96% (pouch skin), 65.30% (tongue) and 60.43% (heart) of RNA-seq reads mapped to the global transcriptome assembly. The lower mapping rate for heart and tongue against the global transcriptome is not unexpected, as reads which map to unannotated transcripts are lost [88]. Alignment of reads from heart and tongue to the genome was higher, with 77.79% and 81.70% of reads mapped, respectively.

## REUSE POTENTIAL

Genomes are valuable tools for wildlife conservation and management [6, 89, 90]. In marsupials, Tasmanian devils [63] and koalas [60] are two examples where genomes have been used to investigate genetic diversity, population structure, adaptation and disease [91–93]. The woylie reference genome is the first genome available for the Potoroidae family of marsupials. Not only will this resource facilitate basic biological research of bettongs and potoroos, but also provide a tool for population genomics studies of woylies and other species within the Potoroidae family. The woylie reference genome has already been used alongside reduced representation sequencing data of woylie populations across Australia to investigate population structure and inbreeding [28].

Infectious diseases threaten wildlife globally, with devastating consequences, such as chytridiomycosis in amphibians and devil facial tumour disease in Tasmanian devils [94]. Genetic diversity within immune genes is essential for adaptation to new and emerging diseases [95]. The cause of the rapid decline of woylie populations in the Upper Warren region of WA remains unknown, but an unknown disease has been hypothesised [96].

The woylie reference genome will enable characterisation of immune genes, an essential first step in determining genetic diversity within these genomic regions and detecting pathogen-driven signatures of selection. Our current understanding of woylie immune genes is extremely limited. The long-read sequencing used to generate the woylie reference genome will enable characterisation of complex immune gene families, such as the major histocompatibility complex. This immunogenetic information will be essential for determining the health of existing populations and mitigating potential future disease outbreaks.

## DATA AVAILABILITY

The reference genome and global transcriptome assemblies supporting the results of this article are available through the Amazon Web Services Open Datasets Program [97]. The genome assembly and all raw sequencing reads including the PacBio HiFi reads, 10x linked-reads and RNA-seq reads are available through NCBI under the BioProject accession PRJNA763700. Annotations, alignments and other results are available via the *GigaScience* GigaDB repository [98].

## DECLARATIONS
## LIST OF ABBREVIATIONS

BLAST: Basic Local Alignment Search Tool; bp: base pair; Benchmarking Universal Single-Copy Orthologs (BUSCO); BP: biological process (BP), GO: Gene Ontology; Gbp: gigabase pair; HMW: high molecular weight; IUCN: International Union for Conservation of Nature; LINE: long interspersed nuclear element; Mbp: megabase pair; NCBI: National Center for Biotechnology Information; NSW: New South Wales; PE: paired end; SINE: short interspersed nuclear element; SA: South Australia; WA: Western Australia.

## ETHICAL APPROVAL

All samples were collected under the Western Australian Government Department of Biodiversity, Conservation and Attractions animal ethics 2018-22F and scientific licence number NSW DPIE SL101204.

## CONSENT FOR PUBLICATION

Not applicable.

## COMPETING INTERESTS

The authors declare that they have no competing interests.

## FUNDING

This work has been funded by the Australian Research Council Centre of Excellence for Innovations in Peptide and Protein Science (CE200100012) and Discovery Project (DP180102465), and the Presbyterian Ladies' College Sydney. PS and LS are supported by an Australian postgraduate award.

## AUTHORS' CONTRIBUTIONS

EP conducted the DNA and RNA extractions, the gene ontology analysis and drafted the manuscript. LS, PB and EP assembled and annotated the genome and transcriptomes. KB and CJH designed the study. All authors viewed, commented on and agreed to publication of the manuscript.

## ACKNOWLEDGEMENTS

The authors would like to acknowledge Adrian Wayne (WA DBCA) and Anke Seidlitz (WA DBCA) for providing samples.

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
