## [Reviewer Report]

Comments on revised manuscriptThe authors addressed all my assembly-related comments in sufficient manner and provided updates that will benefit the manuscript and data released with it.

---

## [Reviewer Report]

Reviewer name and names of any other individual's who aided in reviewer Parwinder KaurDo you understand and agree to our policy of having open and named reviews, and having your review included with the published papers. (If no, please inform the editor that you cannot review this manuscript.)YesIs the language of sufficient quality?YesPlease add additional comments on language quality to clarify if needed
Are all data available and do they match the descriptions in the paper? YesAdditional CommentsAre the data and metadata consistent with relevant minimum information or reporting standards? See GigaDB checklists for examples <a href="http://gigadb.org/site/guide" target="_blank">http://gigadb.org/site/guide</a>YesAdditional CommentsIs the data acquisition clear, complete and methodologically sound?YesAdditional CommentsIs there sufficient detail in the methods and data-processing steps to allow reproduction?YesAdditional CommentsIs there sufficient data validation and statistical analyses of data quality? YesAdditional CommentsIs the validation suitable for this type of data?YesAdditional CommentsIs there sufficient information for others to reuse this dataset or integrate it with other data?YesAdditional CommentsAny Additional Overall Comments to the AuthorRecommendationAccept

---

## [Reviewer Report]

Reviewer name and names of any other individual's who aided in reviewer Walter WolfsbergerDo you understand and agree to our policy of having open and named reviews, and having your review included with the published papers. (If no, please inform the editor that you cannot review this manuscript.)YesIs the language of sufficient quality?YesPlease add additional comments on language quality to clarify if needed
Are all data available and do they match the descriptions in the paper? YesAdditional CommentsThe data on the FTP server available to me covers everything mentioned in the paper.Are the data and metadata consistent with relevant minimum information or reporting standards? See GigaDB checklists for examples <a href="http://gigadb.org/site/guide" target="_blank">http://gigadb.org/site/guide</a>YesAdditional CommentsIs the data acquisition clear, complete and methodologically sound?YesAdditional CommentsIs there sufficient detail in the methods and data-processing steps to allow reproduction?YesAdditional CommentsIs there sufficient data validation and statistical analyses of data quality? YesAdditional CommentsIs the validation suitable for this type of data?YesAdditional CommentsIs there sufficient information for others to reuse this dataset or integrate it with other data?YesAdditional Comments1) 
The submission body and the table 1 state the following assembly stats of the genome assembly that seem to indicate some potential issues:
Genome size (Gb) 3.39
No. scaffolds 1,116
No contigs 3,016
Scaffold N50 (Mb) - 6.94
Contig N50 (Mb)- 1.99
The main issue here for me lies in Scaffold N50 in relation to other parameters, when in comparison with the assemblies using similar methodological approach.
This can either be good or bad, as these numbers might indicate an issue during scaffolding, or presence of long top assembly scaffolds (which is great). I believe, that the submission would significantly benefit if this information is mentioned and discussed.Any Additional Overall Comments to the AuthorThe approach used to generate the assembly seems to utilize 10x PE sequences to scaffold the assembly. There are hybrid assembly approaches available, that leverage short reads to improve the assembly quality, given the slightly limited coverage of PacBio HiFi reads(approx. 12x).RecommendationMinor Revision

---

## [Reviewer Report]

Reviewer name and names of any other individual's who aided in reviewer Qiye LiDo you understand and agree to our policy of having open and named reviews, and having your review included with the published papers. (If no, please inform the editor that you cannot review this manuscript.)YesIs the language of sufficient quality?YesPlease add additional comments on language quality to clarify if needed
Are all data available and do they match the descriptions in the paper? NoAdditional CommentsThe available of all raw sequencing data generated in this study are not stated. And it would be appreciated if the authors could provide a table summarizing all the sequencing data generated in this study. Are the data and metadata consistent with relevant minimum information or reporting standards? See GigaDB checklists for examples <a href="http://gigadb.org/site/guide" target="_blank">http://gigadb.org/site/guide</a>YesAdditional CommentsIs the data acquisition clear, complete and methodologically sound?YesAdditional CommentsCould you also provide the gender information for woy03?Is there sufficient detail in the methods and data-processing steps to allow reproduction?NoAdditional CommentsL145-146: It is unclear how the authors determined full-length protein-coding genes by BLAST against the Swiss-Prot non-redundant database. It would be appreciated if the authors could provide more details here.

L183: The authors indicated that 15,904 of the 24,655 protein-coding genes were supported by mRNA evidence and 1,309 by protein evidence. Does the mRNA evidence come from the RNA-seq data? Where does the protein evidence come from? Is there sufficient data validation and statistical analyses of data quality? NoAdditional CommentsL233: Contaminating sequences in the reference genome are noteworthy, as the DNA for genome sequencing was extracted from wild animals that were dead before sampling. However, I would say high mapping rates did not necessarily represent low contaminating DNA, as the contaminating DNA (e.g. from bacteria), if exists in your dataset, might have been assembled as part of the woylie reference genome. It is unclear if the authors have submitted the genome to NCBI. If so, I think they should have got a report about contamination from NCBI.

It would be appreciated if the authors could provide some more statistics for protein-coding genes (e.g. Mean gene size, Mean exon number per gene, Mean exon length and Mean intron length) and compare these metrics to other marsupials. This will be helpful to judge the quality of the gene models.Is the validation suitable for this type of data?YesAdditional CommentsIs there sufficient information for others to reuse this dataset or integrate it with other data?YesAdditional CommentsAny Additional Overall Comments to the AuthorRecommendationMinor Revision